# Determinants of Successful Aging in a Cohort of Filipino Women

**DOI:** 10.3390/geriatrics4010012

**Published:** 2019-01-11

**Authors:** Emma Tzioumis, Josephine Avila, Linda S. Adair

**Affiliations:** 1Department of Nutrition, Gillings School of Global Public Health, University of North Carolina at Chapel Hill, Chapel Hill, NC 27599, USA; linda_adair@unc.edu; 2Office of Population Studies Foundation, Inc., University of San Carlos, Cebu City 6000, Philippines; jlavila2001@yahoo.com; 3Department of Graduate Architecture, School of Architecture, Fine Arts and Design, University of San Carlos, Cebu City 6000, Philippines; 4Carolina Population Center, University of North Carolina at Chapel Hill, Chapel Hill, NC 27516, USA

**Keywords:** successful aging, self-rated health, low- and middle-income countries, women

## Abstract

This study describes a multidimensional measure of successful aging (SA) and examines the relationship with chronic disease status and self-reported health. Using data from the 2015 Cebu Longitudinal Health and Nutrition Survey of 1568 Filipino women, we created a four domain measure of SA (physiological, mental health, cognitive, sociological). We explored age-stratified associations of each domain and total SA with various health behaviors, chronic disease status, and correlations with self-reported health measures. Both age groups reported aging well, but younger women had higher mean SA scores. Association patterns between domain and total SA and sociodemographic and health behaviors were similar across age groups. Physiological score was associated with hypertension for all ages, and with diabetes in younger women. Total SA was moderately correlated with self-reported health measures. Participants reported aging successfully despite chronic disease status. Future studies should use a multidimensional definition of SA which incorporates elders’ perspective.

## 1. Introduction

The world’s population is aging at a rapid pace, with the number of people over 60 years expected to more than double by 2050 [1]. This demographic transition is especially relevant in low- and middle-income countries, where 80% of the world’s population over 60 years is expected to live by 2050 [2]. In the Philippines, life expectancy is increasing and the number of adults over 50 years will increase from 20 million to 36.5 million in 2040 [3]. The toll of chronic and degenerative diseases will be felt by increasingly younger generations, and rising health care costs will follow. Therefore, identifying factors related to successful aging (SA) will be of great social and public health benefit.

As the world’s age demographics evolve, so does the field of SA. Early studies exploring SA used single dimensional constructs, but recognizing the complex nature of SA transitioned to multidimensional models. One of the most influential and commonly cited models of SA was proposed by Rowe and Kahn [4]. They described SA as consisting of three components: (1) absence of disease or disability, (2) high cognitive and physical function, and (3) active engagement with life. However, this model is has been criticized for its biomedical focus, lack of elders’ perspective, and the high value it unrealistically places on the absence of disease [5,6,7]. Kahn later discussed the complementary nature of various aging theories and that SA is not being without disease or disability [8].

Nevertheless, the Rowe and Kahn model served to generate interest in the complex nature of the aging process, and researchers have since expanded it in many forms. Over time, SA has been interpreted in so many ways that it is widely accepted that there is no standard definition. In a review by Depp and Jeste, 28 studies were identified with 29 different definitions of SA [9]. The components most commonly included in definitions of SA were disability and/or physical functioning, cognitive functioning, social and productive functioning, and life-satisfaction.

Lack of a consistent definition has practical implications, making it hard to draw comparisons. A variable might be included as part of the SA definition in one study, while another study may consider that same variable to be an explanatory factor. Prevalence of SA varies greatly across studies, ranging from 0.4% [10] to 95% [11]. Additionally, most of the research to date has been conducted in Western, industrialized countries, with little work conducted in Asia. More research is also needed in the Asian context, where the concept of aging may differ from the West, with a stronger emphasis on family, staying productive through caring engagement, sense of community, and spirituality as compared to being more goal-oriented in Western societies [12,13,14,15]. Among community-dwelling older adults in Taiwan, SA prevalence was 26.5% when assessed with the Rowe and Kahn model [16]. Ng et al. reported a prevalence of 28.6% in community-living Chinese elderly in Singapore [17]. In a sample of older Korean adults, the mean score for SA was higher than the median (mean 64.3, median 57) [14]. Low prevalence of SA across the world may be due to the emphasis on disease-free state in the Rowe and Kahn definition, as well as a lack of consideration for the cultural context.

The theoretical and empirical variability has prompted calls for increased inclusion of non-biomedical constructs [5] and inclusion of subjective responses from the older adults themselves [7,18]. To this end, we expand upon Young’s definition of SA “as a state wherein an individual is able to make good use of psychological and social potentials to compensate for physiological limitations to achieve a personally satisfying quality of life and a sense of fulfillment even in the context of disease and disability” [19]. The model conceptualizes three domains of SA: (1) physiological (disease and impairment), (2) psychological (emotional vitality), and (3) sociological (engaging with life and spirituality).

We further develop the Young model into four components, separating the psychological domain into distinct mental health and cognitive function domains, to describe SA in a cohort of older Filipino women. We do not include chronic or degenerative diseases in our SA definition, but rather explore the relationship of SA with chronic disease status. Finally, we compare our objective model of SA with three self-reported measures.

## 2. Materials and Methods

### 2.1. Data Source and Study Population

Data come from the Cebu Longitudinal Health and Nutrition Survey (CLHNS), a community-based study of women and their index children followed since 1983. The original participants included all pregnant women in 33 randomly selected communities of Metro Cebu, who gave birth between 1 May 1983, and 30 April 1984 (n = 3327). Study details are described in detail elsewhere [20]. The present study utilizes the most recent round of survey data, collected in 2015-6, which enrolled 1568 mothers (47% of the original cohort). The study was conducted in accordance with the Declaration of Helsinki, and the protocol was approved by Institutional Review Boards at the University of North Carolina at Chapel Hill (IRB #05-1422 and 11-0064) and the University of San Carlos, Cebu, Philippines.

### 2.2. Dependent Variable

Successful aging was defined as a summary score of four domains of health: (1) physiological, (2) cognitive function, (3) mental health, and (4) sociological. Domain 1 was derived from four items; domains 2 and 3 were derived from two items each, and domain 4 was derived from five items. All domains were scaled to have a maximum score of four. Summed, the four domains form a continuous variable representing total SA with a range of 0 to 16, with higher scores indicative of greater SA.

Physiological domain. High scores in the physiological domain represent good physical functioning and absence of disability based on activities of daily living (ADL) [21], instrumental activities of daily living (IADL) [22], physical limitations, and pain. High functioning ADL was defined as being able to perform the following tasks independently: getting dressed, eating, lying down, showering, standing up, and toileting. High functioning IADL was defined as being able to perform the following tasks independently: shopping, preparing food, using transportation, and accounting for money. High functioning physical functioning was defined as reporting no limitations performing the following: household chores, childcare, standing for two hours, walking 100 m, walking 1km, climbing flight of stairs, and carrying a weight of 5 kilos. Pain was self-reported with reference to the previous 30 days (yes/no).

Mental health domain. High scores in the mental health domain represent low levels of depressive symptoms and stress, defined respectively as a score of ≤22 on the Center for Epidemiologic Studies Depression Scale Revised (CESD-R) [23], and a score of ≤20 on the Cohen Perceived Stress Scale [24].

Cognitive function domain. High scores represent absence of cognitive impairment and signs of dementia defined respectively as a score <25 on the Mini-Mental State Examination (MMSE) [25] and a perfect score on the Clock Drawing Test (CDT) [26].

Sociological domain. High scores represent spirituality, active engagement, productive engagement, caring engagement, and support. Spirituality was assessed by the question “How important is it to have religious beliefs in your life?”. Responses of “very important” received a score of 1. Active engagement was defined as being a member of least one community organization (e.g., civic organization, cooperative, women’s group). Caring engagement was defined as self-identifying as being “mainly responsible” for the care of a child or elderly person. Productive engagement was defined as being employed in either paid or unpaid work. Having support was defined as a response of “yes” to the question “Are there people outside of your household with whom you can discuss your personal problems, worries, good news, secrets, or plans in life?”

### 2.3. Independent Variables

Sociodemographic data included age (years), marital status (never, married, separated/divorced, widowed), education (less than primary, primary, some secondary, secondary or more), household composition (single person, one nuclear family, horizontally and/or vertically extended family, multi-nuclear family), and employment status (yes, no). Urbanicity was defined using a community level composite score, with the maximum value of 70 representing the most urban community [27]. Socioeconomic status was a score derived from a tetrachoric factor analysis of reported household assets, using a varimax orthogonal rotation. Two factors with eigenvalues >1.5 were retained: factor 1 represents urban assets (e.g., car, appliances, computer) and factor 2 represents rural assets (e.g., animals, land). Higher values represent higher wealth. A binary variable represents having health insurance (Philhealth and/or private insurance versus none). Health behaviors included smoking (current versus non-smoker and ex-smoker), and alcohol (drank at least one alcoholic drink weekly versus less frequent consumption).

Measures of chronic disease status included diabetes, hypertension, abdominal adiposity, and overweight/obesity. Capillary blood samples were collected and glycated hemoglobin (HbA1C) was immediately measured. Diabetes was defined as HbA1C of >6.5. Blood pressure (BP) was measured with OMRON digital BP devices. Hypertension was defined as systolic BP >140 and/or diastolic >90 mm Hg or taking hypertension medication. Height and weight were measured in home using a portable stadiometer and scale. Overweight was defined as body mass index (BMI) of ≥25 kg/m^2^.

Women reported their overall health general health, quality of life, and satisfaction with social support. We used binary variables to represent “good” or “excellent” responses about general health; a “good” or “very good” rating of quality of life, and “very satisfied” or “satisfied” responses to questions about support from family and friends.

### 2.4. Statistical Analysis

Descriptive statistics for participant characteristics were calculated for the full sample and then separately for those <60 versus 60+ years of age. Due to the non-parametric distribution of the domain scores, the Kruskal-Wallis test was used to compare the means of the domain and total scores across age categories. Univariate analyses were used to explore associations of sociodemographics, health behaviors, and chronic disease status with domains and the total SA score. Relevant variables were retained in a multivariate analysis, stratified by age (<60 years vs. ≥60 years). Within the age strata we further assessed associations with finer age categories. Finally, age-stratified correlations between the binary self-reported health assessments and ordinal SA domain scores were analyzed using polychoric correlation, and for the continuous total SA score with polyserial correlation. Both used the likelihood ratio test. All analyses were conducted in Stata version 14 [28].

## 3. Results

Sample characteristics are shown in Table 1. Age ranged from 46 to 79 years, and 38% of women were ≥60 years. A majority of women were married, and nearly half resided in a multi-nuclear household. Women <60 years were slightly more urban and reported higher urban assets than women ≥60 years. Most women were employed (with or without pay), with a higher proportion of women <60 years being employed than women ≥60 years. Conversely, 72.5% of women ≥60 years had health insurance compared to 43.8% of women <60 years. Many women reported some level of alcohol and cigarette consumption, although amounts consumed or smoked were low. Hypertension, overweight and obesity and excess abdominal adiposity were highly prevalent, with higher adiposity among those <60 years, but higher prevalence of diabetes and hypertension among those ≥60 years.

Figure 1 illustrates mean domain and total SA scores by age group. Women <60 years had a higher mean total SA score and higher domain scores except for mental health.

Table 2 shows the pattern of significant age-stratified bivariate associations of the component domains with sociodemographic and behavioral factors (full results are included in Appendix A). Shaded cells indicate associations with *p* < 0.05. In the ≥60 years group, women 60–65 years had higher scores than those ≥60 years across all domains. In the <60 group, an age gradient was only apparent for the sociological domain. Married women had higher cognitive domain scores than widows, and in the ≥60 years group had higher physiological scores. Across both age groups, higher education was associated with higher mental health, cognitive and sociological scores, but education was unrelated to the physiological domain. Living in a less urbanized community was associated with higher physiological domain scores for all women, and with better cognitive domain scores for those ≥60 years. In both age groups, higher household SES and having health insurance were associated with higher cognitive scores and with higher sociological scores in the <60 years group. Living in a household with extended or multi-nuclear family was associated with higher sociological scores in both age groups. Being employed was associated with higher physiological and mental health scores for all women.

The age-stratified multivariable regression models for each domain (Appendix A) showed that among women <60, current employment was the only variable that remained significantly associated with the physiological and mental health domains. In women ≥60, mental health scores were higher in women 60–65 vs. ≥65 years, and better physiological scores were related to younger age, living in the least urbanized communities, not living alone, and being employed.

In women <60, cognitive domain scores were not different in those <50 vs. 50–60 years. In the <60 age group, higher scores were most strongly related to higher maternal education, higher urban SES, lower rural SES, being currently employed, and having health insurance. Among older women, those 60–65 had higher cognitive scores than those >65 years. Consistent with the younger age group, women with higher education and those currently employed had higher cognitive domain scores. In the sociological domain, among women <60, higher sociological domain scores were associated with younger age, higher education, living in an extended family, and having health insurance. In addition, alcohol consumption and not smoking were related to higher scores. Among women ≥60 years, higher education and living in an extended family were the only predictors of higher sociological scores.

Results of univariate, unadjusted, and adjusted age-stratified models assessing the association between four chronic diseases and the component domains and total SA scores are reported in Table 3. In the univariate analysis, diabetes and hypertension were associated with lower physiological scores for women 60 years. Diabetes, hypertension, high WC, and overweight/obesity were associated with lower physiological scores but higher cognitive function scores in women ≥60 years. The diabetes and hypertension associations with physiological score in women under 60 held in the unadjusted model. In women ≥60 years, the association between hypertension and the physiological score and between diabetes and the cognitive score remained significant. After adjusting for sociodemographic and behavior variables, the associations between hypertension and physiological score remained for all women, and for diabetes and physiological score for women <60 years. No relationships between chronic disease status and cognitive score were observed for women <60 years, or between chronic disease status and any of the other domain scores or total score for women in either age group.

The correlations of self-reported measures with domain and total SA scores are shown in Table 4. In women <60 years, total SA was moderately, positively correlated with overall health and quality of life, and the physiological domain was weakly positively correlated with satisfaction with social support. For women ≥60 years, total SA was moderately, positively correlated with all three measures, and the cognitive function domain was weakly positively correlated with quality of life.

## 4. Discussion

Aging is a phenomena that given the time and opportunity, will affect every individual. Understanding the factors that shape and influence the aging process are therefore critical to supporting wellbeing. In our cohort of older Filipino women, we assessed a multi-dimensional model of SA, and as well as four component domains: physiological, mental health, cognitive function, and sociological. Our model does not assume a chronic disease-free state, but instead we explore the relationship between SA and four common chronic diseases. A further goal was to explore the relationship between our measures of SA and self-reported wellbeing.

We first identified sociodemographic and health behavior factors associated with components of SA. In agreement with recent studies in Asian countries, younger age, higher education, and living in households with extended or multinuclear family were associated with SA in our study [15,16,17,29]. Living in more urban areas was negatively associated with physiological domain in younger and older women, as previously reported [30,31]. The built environment, pollution, occupational and traffic hazards, a shift toward the Western diet, and increases in sedentary behavior in urban areas are factors which could contribute to poorer physiological health outcomes. In the univariate model, SES was associated with higher cognitive scores in both age groups and with higher sociological scores in younger women. Interestingly, in the full model, urban assets were associated with higher cognitive scores but rural assets were associated with lower cognitive scores in younger women (the other relationships were no longer significant). This suggests that these asset variables represent two distinct constructs in this application, not simply the accumulation of wealth. Urban assets represent modern items like computers and other technological devices which can be tools to keep the mind sharp, whereas rural assets include animals and land, and may not encourage similar levels of executive functioning. More work is needed to further probe this relationship. In both age groups, employment was positively associated with physiological and mental health domains, and having insurance was positively associated with the cognitive domain. However, we interpret this finding cautiously, considering the possibility of reverse causation due to the healthy worker effect.

We show that SA can occur despite the presence of chronic disease, reinforcing our decision to exclude chronic disease status from our definition of SA. In the CLHNS sample diabetes, hypertension, high WC, and overweight/obesity were not related to most domain or total SA scores, adding support to a growing body of evidence demonstrating SA and chronic disease can coexist within individuals [11,19,32,33]. Hypertension was associated with lower physiological domain score in both age groups, and diabetes was associated with lower physiological domain score in younger women. These results are in line with reviews reporting a significant association between hypertension and diabetes with functional status decline and physical disability [34,35]. Hypertension has been associated with an earlier onset of disability [36], and others report higher SA with lower vascular risk scores, specifically diastolic blood pressure [37]. Diabetes has been shown to be an independent predictor of physical disability [38]. Although the univariate relationship between diabetes and physiological domain was significant for older women, it was not in multivariable models. One possible explanation for the difference in the diabetes relationship between younger and older women is the influence of both age and cohort effects with respect to weight status and diabetes. Women in this sample are experiencing age and secular trends, where excess weight accumulates faster in younger women in recent times. Women who were 50 years old in 2015 weighed on average 4.6 kg more than women who were 50 years in 2002. Similarly, the oldest women in 2015 had the lowest prevalence of diabetes [39]. Therefore it is possible that in contrast to older women, younger women are experiencing a more severe or complex disease state which is not explained by sociodemographic or health behavior factors.

Women in our sample viewed themselves as aging successfully, as evidenced by over 75% of participants reporting positive perceptions of their overall health, quality of life, and social support, despite experiencing disability and chronic disease. An emerging emphasis is being placed on elders’ perception of their own aging [18,40,41], and this supports previous work showing individuals with chronic conditions can rate their health and life satisfaction highly [11,32,42]. The three self-reported measures were moderately correlated with our objective measure of total SA in both age groups, with slightly higher correlation coefficients in the older group. The component domains were not consistently correlated with the self-reported measures, suggesting that no individual component is driving the relationship between our researcher-defined SA and the elder’s perspective. Others report a clearer relationship between subjective and objective measures, possibly due to the use of variables which more accurately capture SA compared to ours. For example, Strawbridge et. al, asked how strongly the participants agreed or disagreed with the statement “I am aging successfully (or aging well)”, and Montross et. al, asked participants to rate their degree of successful aging on a scale from 1 to 10 [11,40]. Since we conducted an analysis of existing data, these variables were not available.

The strengths of this study are that it prospectively followed a large, community-based sample of women and collected detailed data on objective and subjective health measures. We integrated many of the current critiques of the field to develop a multidimensional model of SA which we assessed along a continuum [6,7,9,43]. Possible limitations should be noted. This was a cross-sectional study, precluding us from establishing causal relationships between the factors considered here and our SA outcome. Longitudinal studies are needed to assess how the dynamics of aging change over time. Attrition is a possible limitation in any cohort study. In the case of the CLHNS, women lost to follow up were more educated and lived in more urbanized communities at baseline than the women that remained in 2015. However, the overall sociodemographic profile of the 2015 sample was similar to that at baseline, minimizing effects on generalizability. Additionally, an attrition analysis found little evidence of selection bias due to attrition for two health outcomes included in the present study–systolic blood pressure and depression score [44]. Although we assess our domain and total SA scores across the range of possible scores to capture a fuller picture of aging, each component domain was constructed of dichotomous variables, possibly reducing the granularity of our measures. An additional limitation is that participants in our sample were younger than populations studied in much of the SA research [9,11,18,29]. For the younger women in our sample, a low score may be less reflective of aging processes, and rather of individual or community factors and it is possible that different relationships would emerge as our cohort ages. However, this is still an important group to study, as experiences earlier in life lay the foundation for subsequent experiences and health outcomes [45,46]. Also, the younger age of our cohort minimizes the effect of survivor bias; those who age more successfully by living longer remain in the study sample, while those who age less successfully die earlier.

## 5. Conclusions

A singular, unifying definition is still lacking in the field of SA research. This study reinforces the importance of including both self-reported and researcher defined SA, and gets to the crux of the successful aging debate [45,46]. One’s own perception of aging will reflect the personal cost of disease, which may be affect absenteeism, productivity, or response to treatment. The objective definitions used by researchers will have use in assessing the societal cost of providing health care services and is useful for identifying characteristics of populations at risk and identifying opportunities and interventions to improve SA. Researchers should continue to work toward a consensus for a multidimensional definition of SA that does not assume a chronic disease free state, assesses aging across a continuum, incorporates elders’ perspective, and acknowledges the societal and cultural norms which shape the aging process.

## Figures and Tables

**Figure 1 geriatrics-04-00012-f001:**
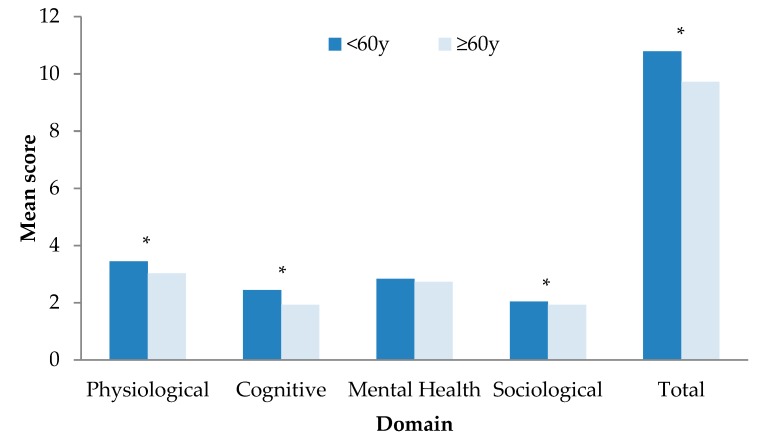
Mean domain and total scores by age group (under 60 years, 60 years and over). Significant differences between age groups are denoted in the figure with an asterisk (*) at *p* < 0.05.

**Table 1 geriatrics-04-00012-t001:** Characteristics of the study participants.

	Under 60 Years		60 Years and Over		Full Sample	
	*n* = 970		*n* = 596		*N* = 1566	
Variables	*n*	%	*n*	%	*N*	%
Age, years ^a^	54.9	3.0	64.9	3.9	58.7	6.0
<50					60	3.8
50–55					413	26.4
55–60					497	31.7
60–65					342	21.8
>65					254	16.2
Marital Status						
Never	5	0.5	0	0	5	0.3
Married	741	76.4	354	59.4	1095	69.9
Widowed	160	16.5	224	37.6	384	24.5
Separated	64	6.6	18	3.0	82	5.2
Education, years ^a^	6.9	2.9	6.5	3.7	6.7	3.2
Urbanicity ^a^	44.4	12.4	42.7	13.3	43.8	12.8
Urban SES ^a^	7.3	5.0	6.7	5.1	7.1	5.1
Rural SES ^a^	6.2	3.6	6.1	3.4	6.2	3.5
Household Composition						
Single person	12	1.2	16	2.7	28	1.8
One nuclear family	274	28.2	149	25.0	423	27.0
Horizontally and/or vertically extended family	218	22.5	156	26.2	374	23.9
Multi-nuclear family	466	48.0	275	46.1	741	47.3
Smoker	293	30.2	197	33.1	490	31.3
Alcohol Consumer	539	55.6	297	49.8	836	53.4
Employed	709	73.1	343	57.6	1052	67.2
Insured	424	43.8	432	72.5	856	54.7
Health status						
Diabetes	121	12.5	102	17.1	223	14.2
Hypertension	444	45.9	328	55.1	772	49.4
High WC	596	61.6	306	51.8	902	57.9
Overweight/obese	515	53.3	247	42.1	762	49.1
Self-reported health						
Overall health (good or excellent)	857	88.4	496	83.2	1353	86.5
Quality of life (good or very good)	744	76.7	462	77.5	1206	77.0
Social support (satisfied or very satisfied)	787	81.1	494	82.9	1281	81.8

*Note*. ^a^ Mean and standard deviation presented for continuous variables.

**Table 2 geriatrics-04-00012-t002:** Heat map of beta coefficients from age-stratified, univariate associations between sociodemographic and behavioral determinants of SA domains.

	Under 60 Years				60 Years and Over			
Variables	Physio Logical	Mental Health	Cog-Nitive	Socio Logical	Physio Logical	Mental Health	Cog-Nitive	Socio Logical
Age, years								
<50	Ref	Ref	Ref	Ref				
50–55	−0.05	0.19	−0.11	−0.24				
55–60	−0.11	0.12	−0.09	−0.29				
60–65					Ref	Ref	Ref	Ref
>65					−0.34	−0.29	−0.70	−0.15
Marital Status								
Never	0.35	−0.47	0.71	−0.13	-^a^	-^a^	-^a^	-^a^
Married	Ref	Ref	Ref	Ref	Ref	Ref	Ref	Ref
Widowed	0.01	−0.15	−0.37	0.00	−0.30	0.04	−0.47	−0.03
Separated	−0.01	−0.06	0.21	−0.14	−0.09	−0.28	0.37	0.33
Education attained								
<Primary	Ref	Ref	Ref	Ref	Ref	Ref	Ref	Ref
Primary	−0.03	−0.03	0.97	0.18	−0.06	0.09	1.00	0.22
Some secondary	−0.10	0.07	1.56	0.17	0.09	0.38	1.58	0.17
Secondary or more	−0.04	0.37	2.01	0.17	0.08	0.68	2.29	0.26
Urbanicity								
Lowest tertile	Ref	Ref	Ref	Ref	Ref	Ref	Ref	Ref
Mid tertile	−0.21	0.01	0.18	−0.08	−0.25	0.05	0.40	−0.05
Highest tertile	−0.29	0.13	0.11	0.05	−0.38	0.02	0.55	0.01
Urban SES								
Lowest tertile	Ref	Ref	Ref	Ref	Ref	Ref	Ref	Ref
Mid tertile	−0.02	0.05	0.74	0.12	0.02	0.08	0.40	−0.08
Highest tertile	−0.07	0.07	1.02	0.15	−0.01	0.12	1.56	0.00
Rural SES								
Lowest tertile	Ref	Ref	Ref	Ref	Ref	Ref	Ref	Ref
Mid tertile	−0.06	0.13	0.13	0.23	−0.07	0.08	0.40	−0.02
Highest tertile	−0.07	0.10	0.40	0.15	0.14	0.22	1.33	0.00
Household composition								
Single person	0.25	0.14	−0.42	−0.25	−0.93	−0.17	−0.50	0.16
One nuclear family	Ref	Ref	Ref	Ref	Ref	Ref	Ref	Ref
Horizontally and/or vertically extended family	−0.07	0.05	−0.14	0.47	−0.24	0.07	−0.01	0.34
Multi-nuclear family	−0.07	−0.08	0.13	0.32	0.01	0.09	0.25	0.22
Smoker	−0.02	0.10	−0.16	−0.10	0.01	0.14	−0.37	0.06
Alcohol Consumer	0.01	−0.01	−0.02	0.10	0.07	0.09	0.10	0.12
Employed	0.13	0.28	0.11	-^b^	0.33	0.29	−0.23	-^b^
Insured	0.02	0.11	0.56	0.15	0.12	0.15	0.38	0.08

^a^ No observations for never married participants over 60 years. ^b^ Not analyzed due to inclusion of employment in construction of sociological domain variable. Shaded cell indicates *p* < 0.05.

**Table 3 geriatrics-04-00012-t003:** Age-stratified associations between chronic diseases and successful aging domain and total scores.

	Under 60 Years			60 Years and Over		
	Univariate	Unadjusted	Adjusted ^a^	Univariate	Unadjusted	Adjusted ^a^
Chronic Disease	β (95% CI)	β (95% CI)	β (95% CI)	β (95% CI)	β (95% CI)	β (95% CI)
Physiological						
Diabetes	−0.31 [−0.47, −0.15]	−0.21 [−0.37, −0.05]	−0.19 [−0.35, −0.03]	−0.36 [−0.60, −0.11]	−0.20 [−0.45, 0.04]	−0.20 [−0.45, 0.05]
Hypertension	−0.18 [−0.28, −0.08]	−0.13 [−0.24, −0.03]	−0.14 [−0.25, −0.03]	−0.35 [−0.54, −0.17]	−0.30 [−0.49, −0.11]	−0.30 [−0.49, −0.11]
High WC	−0.06 [−0.17, 0.04]	−0.10 [−0.26, 0.06]	−0.08 [−0.24, 0.09]	−0.28 [−0.46, −0.09]	−0.10 [−0.36, 0.17]	−0.08 [−0.35, 0.19]
OW/OB	−0.02 [−0.13, 0.08]	0.07 [−0.08, 0.23]	0.08 [−0.08, 0.24]	−0.25 [−0.43, −0.07]	−0.09 [−0.36, 0.17]	−0.10 [−0.37, 0.17]
Cognitive						
Diabetes	0.25 [−0.05, 0.55]	0.23 [−0.08, 0.55]	0.09 [−0.20, 0.37]	0.55 [0.18, 0.91]	0.43 [0.06, 0.81]	0.10 [−0.24, 0.43]
Hypertension	−0.03 [−0.23, 0.18]	−0.09 [−0.30, 0.12]	−0.11 [−0.30, 0.08]	0.38 [0.10, 0.65]	0.17 [−0.11, 0.46]	0.04 [−0.22, 0.29]
High WC	0.19 [−0.02, 0.39]	0.15 [−0.17, 0.47]	0.00 [−0.29, 0.29]	0.59 [0.32, 0.87]	0.29 [−0.12, 0.69]	0.14 [−0.21, 0.50]
OW/OB	0.16 [−0.04, 0.36]	0.06 [−0.25, 0.38]	0.04 [−0.25, 0.32]	0.58 [0.30, 0.86]	0.30 [−0.10, 0.70]	0.15 [−0.20, 0.50]
Mental Health						
Diabetes	−0.02 [−0.26, 0.23]	0.02 [−0.23, 0.27]	−0.04 [−0.30, 0.22]	−0.10 [−0.38, 0.18]	−0.03 [−0.32, 0.27]	−0.07 [−0.37, 0.22]
Hypertension	−0.09 [−0.26, 0.07]	−0.07 [−0.24, 0.10]	−0.05 [−0.22, 0.13]	−0.12 [−0.34, 0.09]	−0.13 [−0.35, 0.09]	−0.11 [−0.34, 0.11]
High WC	−0.10 [−0.27, 0.07]	−0.15 [−0.41, 0.11]	−0.17 [−0.43, 0.09]	−0.02 [−0.23, 0.19]	−0.16 [−0.48, 0.15]	−0.16 [−0.48, 0.16]
OW/OB	−0.04 [−0.21, 0.12]	0.08 [−0.17, 0.33]	0.05 [−0.21, 0.30]	0.10 [−0.12, 0.31]	0.25 [−0.06, 0.57]	0.16 [−0.15, 0.48]
Sociological						
Diabetes	−0.06 [−0.22, 0.10]	−0.05 [−0.22, 0.11]	−0.06 [−0.22, 0.11]	0.05 [−0.15, 0.25]	0.10 [−0.10, 0.31]	0.07 [−0.14, 0.28]
Hypertension	0.00 [−0.11, 0.11]	0.00 [−0.11, 0.11]	0.00 [−0.11, 0.11]	−0.03 [−0.18, 0.12]	−0.07 [−0.22, 0.09]	−0.06 [−0.22, 0.09]
High WC	0.07 [−0.04, 0.18]	−0.02 [−0.19, 0.15]	−0.04 [−0.20, 0.13]	0.04 [−0.11, 0.19]	−0.02 [−0.24, 0.20]	−0.03 [−0.26, 0.19]
OW/OB	0.09 [−0.01, 0.20]	0.10 [−0.07, 0.26]	0.11 [−0.06, 0.27]	0.07 [−0.08, 0.22]	0.10 [−0.12, 0.32]	0.10 [−0.12, 0.32]
Total Score						
Diabetes	−0.15 [−0.64, 0.33]	−0.02 [−0.52, 0.48]	−0.25 [−0.72, 0.21]	0.24 [−0.41, 0.89]	0.38 [−0.28, 1.04]	−0.02 [−0.63, 0.59]
Hypertension	−0.30 [−0.63, 0.02]	−0.31 [−0.64, 0.03]	−0.27 [−0.58, 0.04]	−0.14 [−0.63, 0.35]	−0.29 [−0.79, 0.21]	−0.34 [−0.80, 0.12]
High WC	0.09 [−0.24, 0.42]	−0.09 [−0.60, 0.42]	−0.24 [−0.71, 0.23]	0.24 [−0.24, 0.72]	−0.06 [−0.76, 0.65]	−0.16 [−0.81, 0.48]
OW/OB	0.17 [−0.15, 0.49]	0.28 [−0.21, 0.78]	0.19 [−0.27, 0.65]	0.36 [−0.12, 0.85]	0.46 [−0.25, 1.16]	0.16 [−0.49, 0.80]

Note. 95% CI = 95% confidence interval; WC = waist circumference; OW/OB = overweight or obese. ^a^ Adjusted for sociodemographic and health behavior variables.

**Table 4 geriatrics-04-00012-t004:** Age-stratified correlations of positive self-reported health measures and objective SA domain and total scores.

Domain	Under 60 Years			60 Years and Over		
	Overall Health	Quality of Life	Satisfaction with Social Support	Overall Health	Quality of Life	Satisfaction with Social Support
Physiological	0.55	0.23	0.09 *	0.66	0.35	0.14
Mental health	0.19	0.27	0.12	0.36	0.31	0.15
Cognitive function	0.09	−0.04	−0.08	0.01	0.04 *	0.11
Sociological	0.00	−0.11	−0.12	0.10	−0.14	0.00
Total SA	0.22 *	0.14 *	0.01	0.36 *	0.20 *	0.15 *

Self-reported health measures positively defined as: good or excellent (overall health), good or very good (quality of life), and satisfied or very satisfied (satisfaction with social support). * *p* < 0.05.

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
