# Peer review of "Determinants of Successful Aging in a Cohort of Filipino Women"

_geriatrics, 2019, doi:10.3390/geriatrics4010012_

Round 1
Reviewer 1 Report
The paper is about an interesting topic and it is well written.
I suggest just some minor changes. First: in order to describe the role of disability and age-related diseases in Rowe and Kahn model, I suggest to quote also "Kahn R.L., Successful Ageing: Myth or Reality The 2004 Leon and Josephine Winkelman LectureUniversity of Michigan School of Social Work, March 20, 2004". In this paper Kahn described have well described his position that successful ageing is not being without disorders or diseases or disabilities.
I also suggest to better describe why the choose to study only women.
Author Response
Response to Reviewer 1 Comments
Point 1: In order to describe the role of disability and age-related diseases in Rowe and Kahn model, I suggest to quote also "Kahn R.L., Successful Ageing: Myth or Reality The 2004 Leon and Josephine Winkelman LectureUniversity of Michigan School of Social Work, March 20, 2004". In this paper Kahn described have well described his position that successful ageing is not being without disorders or diseases or disabilities.
Response 1: Thank you for the suggested citation. We have added it in the introduction, at line 41.
Point 2: I also suggest to better describe why the choose to study only women.
Response 2: The Cebu Longitudinal Health and Nutrition Survey (CLHNS) was originally designed as an infant feeding study. The sample included all women in the metro Cebu study communities who were pregnant at baseline in 1983. While the children (male and female) of these women continue to be followed, the children are not yet old enough to be included in a study on successful aging. Therefore, this paper only analyzes the mothers of the children. We do not have detailed health data on the fathers.

Reviewer 2 Report
This is a well written manuscript and that provides some potentially important insights to promote SA but you might mention a weaknesses from your 2011 publication - Cohort Profile: The Cebu Longitudinal Health and Nutrition Survey (2011) -- Attrition is a concern, especially given the high mobility of the young adult population. Migration of the more educated, urban segment of the original cohort has left us with a sample that is no longer representative of the population from which it was drawn.
Author Response
Response to Reviewer 2 Comments
Point 1: You might mention a weaknesses from your 2011 publication - Cohort Profile: The Cebu Longitudinal Health and Nutrition Survey (2011) -- Attrition is a concern, especially given the high mobility of the young adult population. Migration of the more educated, urban segment of the original cohort has left us with a sample that is no longer representative of the population from which it was drawn.
Response 1: Thank you for the point. We have included information from a forthcoming attrition analysis in the discussion section, at line 294 (reference 46: L. S. Adair, T. L. Perez, and J. B. Borja, “The life history of a cohort study: Attrition in the Cebu Longitudinal Health and Nutrition Survey,” Philipp. Popul. Rev., Forthcoming.) Specifically, we note, based on a more up-to-date analysis, that “women lost to follow up were more educated and lived in more urbanized communities at baseline than the women that remained in 2015. However, the overall sociodemographic profile of the 2015 sample was similar to that at baseline, minimizing effects on generalizability. Additionally, an attrition analysis found little evidence of selection bias due to attrition for two health outcomes included in the present study – systolic blood pressure and depression score [43]”.
